# Effect of Palm Kernel Cake Supplementation on Voluntary Feed Intake, In Situ Rumen Degradability and Performance in Buffaloes in the Eastern Amazon

**DOI:** 10.3390/ani13050934

**Published:** 2023-03-04

**Authors:** João Maria do Amaral-Júnior, Eziquiel de Morais, Alyne Cristina Sodré Lima, Lucieta Guerreiro Martorano, Benjamim de Souza Nahúm, Luciano Fernandes Sousa, José de Brito Lourenço-Júnior, Thomaz Cyro Guimarães de Carvalho Rodrigues, Jamile Andréa Rodrigues da Silva, Artur Luiz da Costa Silva, André Guimarães Maciel e Silva

**Affiliations:** 1Federal Institute of Amapá (IFAP), Porto Grande 68997-000, AP, Brazil; 2Federal Institute of Pará (IFPA), Castanhal 68741-400, PA, Brazil; 3Embrapa Eastern Amazon, Santarém 68020-640, PA, Brazil; 4Embrapa Eastern Amazon, Belém 66095-903, PA, Brazil; 5Department of Animal Science, Federal University of Tocantins (UFT), Araguaína 77804-970, TO, Brazil; 6Postgraduate Program in Animal Science (PPGCAN), Institute of Veterinary Medicine, Federal University of Pará (UFPA), Belém 66075-110, PA, Brazil; 7Institute of Animal Health and Production, Federal Rural University of the Amazon (UFRA), Belém 66077-580, PA, Brazil; 8Postgraduate Program in Animal Science (PPGCAN), Institute of Veterinary Medicine, Federal University of Pará (UFPA), Castanhal 68746-630, PA, Brazil

**Keywords:** co-products, oilseed, nutrition, buffaloes, weight gain

## Abstract

**Simple Summary:**

The destination of agro-industrial residues and deforestation to form new pasture areas are urgent concerns in the Amazon region. Diverse studies have evaluated the inclusion of these residues in ruminant diets and have enabled the intensification of animal production systems. Palm kernel cake (PKC), for example, is generated after oil extraction for the food and cosmetics industry, but it still has crude protein and metabolizable energy levels of around 17% and 7.6 MJ/kg in the matter dry (DM), respectively. This research aims to evaluate the inclusion of a PKC-based supplementation for grazing animals on feed intake, degradability, and productive performance of buffaloes in the eastern Amazon. Although some changes in nutrient intake and degradability were observed, the inclusion of the co-product in the diet did not promote changes in productive performance and reduced forage consumption. Thus, the use of PKC up to 1% of body weight in supplements is recommended as an alternative source of nutrients for grazing animals.

**Abstract:**

The objective was to evaluate the effects of palm kernel cake (PKC) supplementation on voluntary feed intake, in situ rumen degradability and performance in the wettest (WS—January to June) and less rainy seasons (LR—July to December) in the eastern Amazon. A total of 52 crossbred buffaloes that were neither lactating nor gestating were used, with 24 for the LR, aged 34 ± 04 months and an initial average weight of 503 ± 48 kg, and 24 for the WS aged 40 ± 04 months with an average weight of 605 ± 56 kg. The four treatments (levels of PKC in relation to body weight) were distributed in a completely randomized design, with 0% (PKC0), 0.25% (PKC0.2), 0.5% (PKC0.5) and 1% (PKC1) with six repetitions. The animals were housed in Marandu grass paddocks, intermittently, with access to water and mineral mixture ad libitum. Degradability was evaluated by the in situ bag technique in four other crossbred buffaloes with rumen cannulae, in a 4 × 4 Latin square (four periods and four treatments). The inclusion of PKC increased supplement consumption and production of ether extracts and reduced the intake of forage and non-fibrous carbohydrates. The dry matter degradability of Marandu grass was not affected; however, the fermentation kinetics in neutral detergent fiber (NDF) differed between the treatments. The co-product dry matter colonization time was greater in PKC1 and the highest effective degradability rates were from PKC0, but the productive performance of the animals was not influenced. Supplementation of buffaloes with PKC is recommended for up to 1% of body weight.

## 1. Introduction

Livestock is seen as an intensifier of the deforestation process in the Amazon. As a result, strategies such as the supplementation of animals in an extensive management system has been improved, aiming to combine improvements in production with better use of available areas [1,2].

In this sense, the use of by-products has proved to be an interesting alternative to corn or soybean meals, since in addition to making use of agro-industry residues, it has low market value and good nutritional potential as a replacement for traditional ingredients [2,3]. Its inclusion in ruminant diets, in addition to reducing environmental impacts, promotes the intensification of animal management systems and can help to increase productivity per area [4,5,6].

Among available co-products, palm kernel cake (PKC) from the dendê palm (*Elaeis guineensis*) stands out, because, in addition to its low cost and increasing availability, it has a dry matter content between 88.11 and 97.7%, ash between 3.01 and 7.82%, ethereal extracts ranging from 5.7 to 13.55%, crude protein between 13.01 and 18.23%, and neutral detergent fiber from 64.09 to 81.85% [7,8,9].

However, PKC has a high content of acid-detergent fiber, which is slowly degraded and can restrict feed intake, degradability, and animal performance [10]. Thus, for better use in animal feed, it is important to know its behavior in the rumen environment, its effect on other ingredients in the diet, and its inclusion limit.

The hypothesis of this study is that there exists a supplementation level of PKC in buffalo diets that can maximize productive performance in pasture-grazed animals. Thus, the objective of this study was to evaluate voluntary feed intake, in situ rumen degradability and productive performance of buffaloes in an intermittent stocking system supplemented with palm kernel cake in the eastern Amazon. 

## 2. Materials and Methods

### 2.1. Ethics Committee and Experiment Location

All procedures with the animals were performed with authorization from the Ethics Committee on the Use of Animals of Embrapa Eastern Amazon, under protocol No. 007/2015.

The experiment was carried out at the Animal Research Unit “Dr. Felisberto Camargo” (01°25′ S and 48°26′ W), belonging to Embrapa Eastern Amazon, municipality of Belém, Pará, Brazil. The climate is Afi in the Köppen classification, A—humid, tropical climate with no cold period and with average monthly temperature of the warmest month above 18 °C; f—tropical forest climate with rainfall relatively abundant throughout the year, and i—the amplitude between the average temperatures of the warmest and coldest months is less than 5 °C with an average rainfall of 3001 mm/year, well distributed over the months. The average annual temperature and relative humidity are 26 °C and 86%, respectively, with the wettest period (WS) between January and June and the least rainy (LR) from July to December. 

### 2.2. Animals and Experimental Design

To evaluate feed intake and performance, the study design used 24 buffaloes, non-lactating and not pregnant, with initial body weight of 503 ± 48 kg and approximately 34 months of age (LR), and another 24 with initial body weight of 605 ± 56 kg and 40 ± 04 months of age (WS), distributed in a completely randomized design, with 4 treatments (PKC levels), 6 repetitions (animals) and repeated measures in time (LR and WS). Inclusion levels were: 0% (control treatment), and 0.25; 0.50 and 1% of body weight (BW). For degradability, 4 buffaloes with rumen cannula, in a 4 × 4 Latin square (4 periods and 4 treatments) were used. 

### 2.3. Experimental Diets and Chemical Analyses

Supplementation was offered once a day, at 08:00 h, in a stable with individual stalls. Furthermore, the animals were kept in 8 paddocks (1.2 ha) of Urochloa brizantha grass (*syn. Brachiaria brizantha*) cv. Marandu (*Marandu grass*), totaling an area of 9.6 ha, and provided with water and mineral mixture ad libitum in an intermittent stocking system, with 4 days of occupation and 28 days of rest.

For greater acceptance of the supplement, the animals underwent a period of adaptation to the diet and management (21 days), and 0.15% of body weight of wheat bran was included in all treatments (Table 1). 

The forage mass was measured using the double sampling method, where destructive estimates were associated with canopy height [11]. Every 90 days, 60 height readings were measured and recorded, at random, in all experimental paddocks, using a graduated cane, and forage samples were collected at ground level at 9 predetermined points (3 of shortest height, 3 medium height and 3 tallest).

In the laboratory, 2 samples composed of whole plants were generated (pool of samples of height), which were dried in an oven (55 °C for 72 h) and weighed again to determine the forage/picket mass and chemical composition.

Subsequently, samples were ground in a Wiley mill using a 1 mm sieve for dry matter (DM; method 934.01), ash (Ash; method 930.05), crude protein (CP; method 981.10) and ether extract (EE; method 920.39) [12]. The content of neutral detergent fiber (NDF) and fiber in acid detergent (ADF) was determined according to [13]. Non-fibrous carbohydrates (NFC) in diets were calculated according to [14], where NFC% = 100% − [CP% + NDF% + EE% + Ash%]. 

Grass samples were collected by simulated grazing (every 90 days), performed by trained professionals, to evaluate the portion consumed by the animals [15].

### 2.4. Voluntary Feed Intake and Performance

The feed intake of the supplement and nutritional components was determined by the difference between the amount contained in the provided feed and the amount contained in the leftovers. Daily, before the new portion of supplement was offered, the leftovers were collected and weighed to determine the dry matter consumption, and this was used to adjust the next portion to allow 15% of leftovers. 

To estimate fecal production (kg DM/feces/day), titanium dioxide (TiO_2_) was used as an external indicator (10 g/animal/day), applied directly into the esophagus through a probe. The indicator was given for 7 days for adaptation purposes to obtain a homogeneous plateau of excretion [16].

The determination of TiO_2_ content was performed by collecting feces, directly from the rectal ampulla, in the 24 animals, during the 2 experimental periods (LR and WS), from the eighth day of administration. The samples were packed in plastic packaging and kept at −20 °C. At the end of the period, the samples from each animal were homogenized, constituting a composite sample (sample/animal) and then pre-dried, ground and packed in flasks. A sample of 0.5 g of feces was digested for 2 h at 400 °C and, soon after, 10 mL of H_2_O_2_ (30%) was slowly added and completed with distilled water until it reached twice the volume. Subsequently, the material was transferred to 100 mL flasks and 3 drops of H_2_O_2_ (30%) were added [17].

Feces, simulated grazing material, supplements and leftovers were analyzed for DM, CP, EE and ash [18], for NDF and ADF by the sequential method [19], and in vitro dry matter digestibility (DIVDM), simulated grazing and PKC [20].

Fecal excretion was calculated using the formula: FE = TiO_2_ (administered (g/day))/TiO_2_ (existing in fecal DM (g/kg)). The feces of each animal were weighed, dried (55 °C ± 5 °C, for 72 h) and ground to pass a 2 mm sieve to determine the indigestible neutral detergent fiber (NDFi) and, later, the material was ground to pass a 1 mm sieve to evaluate the chemical composition. 

To estimate individual forage consumption (FC) (DM), the internal marker (NDFi) was used [21]. The estimation of dry matter consumption was carried out from the Equation:$$\mathrm{FC}(g/\mathrm{dia})=\frac{\left[(\mathrm{EF}(g/\mathrm{day})\;\times \;\mathrm{FCNDFi}(\%)\;÷\;100)\;-\;\mathrm{ISNDFi}\;(g/\mathrm{day})\right]}{\mathrm{CIFOR}\;\mathrm{NDFi}(\%)}\;\times 100$$
where FC = individual forage consumption in g/day; EF = fecal excretion in g/day; CIFNDFi = concentration of NDFi in feces in %; ISFDNi = NDFi intake via supplement in g/day; FCFDNi = NDFi concentration in forage in %. 

To assess the performance, weights were measured at time zero (day 1) and then every 28 days (after 16 h of fasting from solids and liquids). Total weight gain (TWG) was determined by the difference between final and initial weight. The average daily gain (ADG) was calculated by dividing the TWG by the number of days of duration of the experiment for each period (LR and WS).

### 2.5. Rumen Degradability

The rumen in situ bag technique was used to measure rumen degradability [22]. The study design used 4 crossbred buffaloes with ruminal cannulae, and an initial age and average weight of 34 ± 04 months and 503 ± 48 kg, respectively, in a 4 × 4 Latin square experimental design (4 treatments with 4 repetitions) with repeated measures in time.

Buffaloes were supplemented with the same levels of PKC and access to Marandu grass paddocks (Table 2). There were 4 periods of 25 days, with 21 days of adaptation of the animals to the diets and in situ incubation of the feed for a period of 4 days. The incubation times were: 0, 6, 12, 24, 48, 72 and 96 h, except for wheat bran, which was incubated for 48 h, with the bags placed in reverse order for removal at the same time.

For the incubation of Marandu grass, the samples were dried and ground in a knife mill with a 5 mm sieve. PKC and wheat bran (WB) samples were ground to pass a 2 mm sieve [22]. Degradability was determined by the “in situ” technique, using 100% polyamide-nylon unresined bags (14 × 7 cm), with approximately 50-micrometer pores, containing 5 g of sample, at a rate of 20 mg DM/cm^2^.

After removal, the bags were washed in cold water to remove excess rumen content, immersed for 30 min in ice water to stop microbial activity and subsequently frozen. At the end of the process, all bags were washed in a washing machine (5 cycles of 10 min). Unincubated bags containing the same amounts of samples were washed along with incubated bags to represent the zero hours of incubation. 

After washing, the bags were dried in a forced ventilation oven (55 °C for 72 h). The percentage of DM, NDF, ADF and CP by incubation time was calculated by the difference between the incubated and the remainder in the samples, thus generating the degradation curves.

The effects of treatments and incubation times on the dependent variables were evaluated using the model of [23]: *p* = a + b (1 − e − ct), where *p* is the disappearance in time (t); a (fraction soluble); b (degradable fraction); c (rate of degradation of b); and t (time in hours). 

Effective degradability was estimated using the equation ED = a + [(b × c)/(c + k)], where ED is the effective degradability of nutrients; a (rapidly soluble fraction); b (potentially degradable fraction); c (rate of degradation of parameter b); and k (rate of passage of digestate out of the rumen), considered for rates of 0.02 h^−1^, 0.05 h^−1^ and 0.08 h^−1^ [24].

Colonization time was estimated by the equation: CT = − (1/c) × {ln [(a + b − S)/b]}, where a, b and c are the same parameters as in the previous equation and S is the soluble fraction at time zero [25]. In the fibrous fractions (NDF and ADF), the soluble fraction “S” was considered zero.

The nonlinear parameters a, b and c were estimated by the algorithmic procedure of Gauss–Newton.

### 2.6. Statistical Analysis

The data were analyzed by means of analysis of variance and regression, with the degrees of freedom broken down into linear or quadratic effects, the significance of up to 5%, using the PROC GLIMMIX of the Statistical Analysis System—SAS version 9.1 (SAS, 2009), according to the statistical model below:Ŷij = μ + NLi + Ɛij
where Ŷij = value observed in the plot that received treatment i in repetition j; μ = overall average; NLi = fixed effect of oil palm inclusion level i (i = 0, 0.25, 0.5 and 1%); and Ɛij = random experimental error associated with each observation assumed NID ~ (0, σ2).

## 3. Results

### 3.1. Voluntary Feed Intake

Supplement intake (Kg/DM) increased linearly with the inclusion of palm kernel cake, reaching 5.57 and 5.31 kg at the highest inclusion level (WS and LR, respectively), but it was not influenced by season, nor was there any interaction (Table 3). Forage consumption (Kg/DM) was reduced with the inclusion of the co-product and was influenced by the change of season, also with a reduction in consumption; however, there was no interaction. The total DM intake was influenced only by the periods, with a reduction during the LR. 

Supplement consumption (%LW) was influenced only by treatments, reaching 0.9 kg in the treatment with 1% inclusion of palm kernel cake compared to 0.15 kg in the control treatment. On the other hand, forage consumption between treatments and seasons was reduced; however, without interaction. Total consumption was influenced only by the season, being higher in WS, with an average of 1.778 compared to 1.640 in LR.

In relation to metabolic weight, all intakes were influenced by the inclusion of PKC (*p* < 0.05). However, only forage consumption was influenced by the season, with a linear reduction as well as an interaction.

The consumption of CP, NDF, EE and NFC per kg of DM was influenced by the season, increasing in WS (*p* < 0.05). However, only the consumption of EE and NFC was influenced by the treatment, increasing and decreasing, respectively, with the highest levels of PKC inclusion (Table 4).

The seasons also influenced the consumption of all fractions (% BW), being higher in WS (*p* < 0.05). The treatments only influenced the consumption of EE and NFC, with an increase (136.7%) and a decrease (33.5%), respectively.

For feed consumption in relation to metabolic weight, the results followed the same pattern, influencing all fractions (CP, NDF, EE and NFC) with the change of season, being higher in WS, with the inclusion of the PKC supplement only influencing the intake of EE and NFC, with an increase and a decrease, respectively. 

The consumption of NFC reduced linearly in the two seasons and, in the comparison between the two, there was also a reduction (*p* < 0.05). In the WS, the reduction reached 25% and, in the LR, it reached 42.2%, between the 0% and 1.0% treatments (Table 5).

### 3.2. Degradability

The inclusion of palm kernel cake in the diets influenced the in situ degradation of DM and NDF (*p* < 0.05), while the CP degradation equations and curves were similar (*p* > 0.05) (Table 6). The values of the coefficient of determination (R^2^) were greater than 92%, indicating a good fit of the data to the model equation [26].

The degradation of palm kernel cake NDF was different at the TP0.25 level compared to the others (*p* < 0.05), and this level displayed atypical behavior, with a higher soluble fraction (8.66%) and higher TC (6:43 h).

There was no difference in the degradation (DM) of Marandu grass (*p* > 0.05); however, the degradation of NDF and CP differed (*p* < 0.05) (Table 7).

The NDF degradation equations for Marandu grass for the TP0 and TP1 treatments are different (*p* < 0.05), highlighting the values of the soluble fraction (A) (1.17 and 1.71, respectively), which may have contributed to the result of the kinetic parameters of the curves. For CP degradation, Marandu grass equations (PKC treatments) were different from the control treatment (TP0) (*p* < 0.05).

Wheat bran DM and CP degradation were influenced by the inclusion of the palm kernel cake but there was no difference in NDF degradation (*p* > 0.05) (Table 8).

Wheat bran DM ruminal degradation equations were similar in terms of kinetic behavior at supplementation levels TP0.5 and TP1 (*p* > 0.05); however, treatments TP0 and TP0.2 were different (*p* < 0.05). The colonization time of wheat bran was longer in the treatments with palm kernel cake, with a difference of 101 and 140 min between the 0% and 1% treatments for DM and CP, respectively. 

With respect to NDF, since the potentially degradable fraction was the same, and the degradation rates were similar, it can be inferred that the proposed PKC levels do not promote any effect on the degradability of wheat bran fiber.

In the wheat bran CP degradation equations, there was an increase in the potentially degradable fraction, from 55.44% in TP0 to 89.99 in TP1. The highest effective degradability rates were observed in TP0 (86.41%, 79.04% and 73.60%) for the rates of 2%/h^−1^, 5%/h^−1^ and 8%/h^−1^, respectively, as well as the shortest colonization time (0:14 h).

### 3.3. Performance

Palm kernel cake did not influence performance variables (*p* > 0.05) (Table 9). However, the average daily gain ranged from 0.478 to 0.544 kg in LR and from 0.211 to 0.339 kg in WS. The total average gain varied between 86 and 98 kg in the LR and, in the WS, from 26 to 62 kg.

## 4. Discussion

### 4.1. Feed Intake

Concentrated feeds usually have high acceptability by animals, with consumption varying according to the ingredients used [27,28]. Thus, the inclusion of PKC was a determining factor for the increase in supplement intake, regardless of the season.

However, pasture supplementation tends to reduce forage consumption, as it promotes greater nutrient intake and, consequently, increases efficiency in the use of forage energy [29]. Levels of supplementation greater than 0.3% of body weight/day significantly reduces the consumption of pasture grass [30]. This fact contributes to animal performance and allows for more efficient use of available pastures and areas [31]. 

The reduction in dry matter consumption (kg·day^−1^) can also be attributed to the inclusion of the supplement since it is a feed with greater nutritional density. Additionally, the presence of palm kernel cake in the diets increased the levels of ether extract, which was observed at the highest levels of PKC inclusion (TP0.5 and TP1), which can cause a reduction in fiber digestibility [32] (Table 4).

The greater consumption of WS compared to LR provided a greater intake of the NDF, EE and NFC fractions per kg of DM, body weight and metabolic rate. The increase in EE consumption and reduction in NFC intake reflects the change caused by palm kernel cake in the nutritional composition of the diet. This co-product is high in fat and ADF but low in NFC, thus directly influencing the intake of these fractions (Table 2) [7,8,33]. The lower values found in the LR season are probably a reflection of the lower DM consumption (Table 3).

For digestible energy, the reduction possibly occurred by replacing pasture with concentrate, since there was a reduction in roughage consumption with supplementation, resulting in an increase in digestible energy intake.

### 4.2. Degradability

Palm kernel cake, despite its good nutritional composition in protein and energy, raises ADF and lignin levels, promoting changes in the digestive process, such as longer degradation time, microorganism action and passage rate [34,35,36]. These changes may reflect reduced intake, digestibility, and animal performance [10,37].

The influence on fiber degradation, in turn, may have occurred due to the higher consumption of EE by animals (TP1), since the addition of lipids in ruminant diets may have this effect [38,39]. Furthermore, the limitation in the degradation of palm kernel cake is due to the indigestible fraction associated with the endocarp residues present in this ingredient, a highly lignified and extremely hard tissue [40].

Ruminal degradability and digestibility are key measures to consider when determining the nutritive value of new ingredients [41]. As the potentially degradable fraction was the same and the degradation rates were similar, it can be inferred that, regardless of the TP level, there was no effect on wheat bran fiber degradability [42].

The differences in the Marandu grass CP degradation equations may be due to the average increase in TC of 1:37h between the TP0 levels and the treatments with the inclusion of palm kernel cake.

The effect of PKC on the degradation equations also possibly reflects the increase in the ADF and lignin fractions present in the co-product. Lignin, in addition to being indigestible, can make it impossible to digest other fractions, limiting access by microorganisms and causing longer colonization times. This may have occurred, for example, in the protein equations, where there was an average increase in TC of 107 min between the most distant levels.

The influence on wheat bran degradation was probably due to the composition of the palm kernel cake and the changes caused in the rumen environment and digestive processes, such as the increase in colonization time for fiber degradation and the increase in fat.

### 4.3. Performance

The equality observed in the results demonstrates that, despite the variations in consumption and nutritional components, the inclusion of the co-product provides an equivalent performance of the animals, as already observed in goats [9,43] and cattle [10,35,44].

Adequate pasture management conditions, with an average supply of 8 kg of DM/100 kg of BW, above the minimum requirement required by buffaloes for good productive performance, which is 1200 to 1600 kg of DM ha^−1^ [45], and supplementation with PKC may have contributed to the equality in the performance of the animals.

## 5. Conclusions

Supplementation of buffaloes with palm kernel cake reduces forage consumption, thus reducing the necessity of increasing pasture area, and it influences diet degradability but does not interfere with performance. More studies are recommended to improve the use of this co-product as part of a sustainable feeding management strategy.

## Figures and Tables

**Table 1 animals-13-00934-t001:** Chemical composition of the ingredients of the experimental diets.

Items	PKC		WB		MG	
	LR ^1^	WS ^2^	LR	WS	LR	WS
DM ^3^%	90.47	89.30	88.32	89.70	31.10	27.90
CP ^4^ (%DM)	11.12	13.40	15.49	16.28	8.19	8.84
NDFcp ^5^ (%DM)	69.87	59.90	44.19	42.61	68.14	67.19
ADFcp ^6^ (%DM)	48.23	31.40	14.27	13.21	40.55	38.21
Ash (%DM)	4.61	4.26	5.88	5.03	6.76	6.44
EE ^7^ (%DM)	11.64	13.19	3.48	3.19	2.54	2.63
NFC ^8^ (%DM)	2.76	9.25	30.96	32.89	14.37	14.90

PKC = Palm kernel cake; WB = wheat bran; MG = Marandu grass; ^1^ Less rainy period (July to December); ^2^ Wettest period (January to June); ^3^ DM = dry matter; ^4^ CP = crude protein; ^5^ NDFcp = neutral detergent fiber corrected for ash and protein; ^6^ ADFcp = fiber in acid-detergent corrected for ash and protein; ^7^ EE = ether extract; ^8^ NFC = non-fibrous carbohydrates.

**Table 2 animals-13-00934-t002:** Chemical composition of the diets with the inclusion of palm kernel cake supplement based on dry matter.

Items	Supplement Treatments			
	PKC 0	PKC 0.2	PKC 0.5	PKC 1
DM ^1^ (g/kg supplement)	883.2	883.0	899.0	902.0
CP ^2^ (g/kg DM)	154.9	127.0	121.0	117.0
NDFcp ^3^ (g/kg DM)	441.9	602.0	640.0	666.0
ADFcp ^4^ (g/kg DM)	142.7	355.0	405.0	439.0
Ash (g/kg DM)	58.8	50.8	48.9	47.7
EE ^5^ (g/kg DM)	34.8	85.8	97.9	106.0
NFC ^6^ (g/kg DM)	309.6	133.0	91.4	63.6

^1^ Dry matter; ^2^ Crude protein; ^3^ Neutral detergent fiber corrected for ash and protein; ^4^ Fiber in acid-detergent corrected for ash and protein; ^5^ Ether extract; ^6^ Non-fibrous carbohydrates.

**Table 3 animals-13-00934-t003:** Supplement, Marandu forage, and total feed intake by buffaloes in two seasons in the eastern Amazon.

Period	Inclusion PKC (% BW)				Average ^1^	CV	DMS	SEM ^3^	*p*-Value		
	0.0	0.25	0.50	1.0					G	*p*	G × *p*
Average supplement consumption (kg of DM)											
WS ^4^	0.98 d	2.56 c	4.15 b	5.57 a	3.31 A	10.8	0.406	0.142	<0.01	0.08	0.89
LR ^5^	0.91 d	2.40 c	3.93 b	5.31 a	3.14 A						
Average ^2^	0.94 d	2.48 c	4.04 b	5.44 a			0.287	0.100			
Average forage consumption (kg of DM)											
WS	11.12 a	7.97 b	7.07 b	5.73 c	7.98 A	11.2	0.958	0.335	<0.01	<0.01	0.37
LR	9.17 a	7.15 b	5.97 c	4.27 d	6.64 B						
Average	10.15 a	7.56 b	6.52 c	4.99 d			0.677	0.237			
Average of total consumption (kg of DM)											
WS	12.10	10.52	11.22	11.3	11.28 A	9.3	1.147	0.401	0.087	<0.01	0.58
LR	10.08	9.56	9.902	9.57	9.780 B						
Average	11.09	10.04	10.56	10.43			0.811	0.283			
Supplement consumption (% LW)											
WS	0.15 d	0.400 c	0.650 b	0.900 a	0.525 A	1.7	0.010	0.003	<0.01	0.41	0.57
LR	0.15 d	0.400 c	0.650 b	0.908 a	0.527 A						
Average	0.15 d	0.400 c	0.650 b	0.904 a			0.007	0.002			
Forage consumption (% LW)											
WS	1.726 a	1.247 b	1.109 bc	0.931 c	1.253 A	13.3	0.184	0.064	<0.01	<0.01	0.65
LR	1.533 a	1.195 b	0.987 c	0.738 d	1.113 B						
Average	1.629 a	1.221 b	1.048 c	0.834 d			0.130	0.045			
Total consumption (% LW)											
WS	1.876	1.647	1.759	1.831	1.778 A	9.13	0.182	0.063	0.09	<0.01	0.66
LR	1.683	1.595	1.637	1.646	1.640 B						
Average	1.779	1.621	1.698	1.738			0.128	0.045			
Supplement consumption (BW^0.75^)											
WS	0.042 d	0.113 c	0.183 b	0.252 a	0.147 A	3.53	0.003	0.001	<0.01	0.28	0.97
LR	0.041 d	0.111 c	0.181 b	0.250 a	0.146 A						
Average	0.042 d	0.112 c	0.182 b	0.251 a			0.004	0.001			
Forage consumption (BW^0.75^)											
WS	259.6 aA	89.09 bA	54.95 cA	34.89 dA	109.6 A	9.0	10.59	3.706	<0.01	<0.01	<0.01
LR	214.0 aB	79.99 bA	46.40 cA	25.80 dA	91.56 B						
Average	236.8 a	84.54 b	50.68 c	30.34 d			7.490	2.620			
Total Consumption (BW^0.75^)											
WS	45.47 c	50.20 c	58.59 b	67.64 a	55.48 A	11.1	7.03	2.459	<0.01	0.25	0.78
LR	41.67 c	47.20 c	56.34 b	68.58 a	53.45 A						
Average	43.57 d	48.70 c	57.46 b	68.11 a			4.97	1.73			

^1^ Average of the period; ^2^ Average of treatment; ^3^ Standard error of the mean; ^4^ the wettest season; ^5^ the less rainy season; LW = live weight; BW = body weight; G = Treatment; *p* = Period; Equal lowercase letters in the same row means no significant difference. Different lowercase letters in the same row indicate a significant difference. Equal capital letters in the same column mean no significant difference. Different capital letters in the same column indicate a significant difference.

**Table 4 animals-13-00934-t004:** Chemical analysis of consumption as a function of increasing levels of palm kernel cake by buffaloes in two seasons in the eastern Amazon.

Items	Inclusion PKC				Periods		Effects			R^2^	SEM
	0.0	0.25	0.5	1.0	LR	WS	TREAT	PER	TREAT*PER		
	Consumption (kg of DM)										
CP	1.017	0.981	1.085	1.120	0.929 B	1.172 A	NS	*	NS	-	0.005
NDF	7.266	6.516	6.824	6.747	6.498 B	7.179 A	NS	*	NS	-	0.045
EE	0.293 d	0.418 c	0.587 b	0.72 a	0.464 B	0.546 A	*	*	NS	96.01	0.019
NFC	1.785 a	1.496 b	1.436 b	1.240 c	1.274 B	1.705 A	*	*	NS	90.12	0.058
	Consumption (% BW)										
CP	0.169	0.159	0.176	0.179	0.157 B	0.185 A	NS	*	NS	-	0.011
NDF	1.213	1.058	1.113	1.085	1.097 B	1.137 A	NS	*	NS	-	0.074
EE	0.049 d	0.067 c	0.095 b	0.116 a	0.078 B	0.086 A	*	*	NS	95.74	0.006
NFC	0.298 a	0.242 b	0.233 b	0.20 b	0.215 B	0.270 A	*	*	NS	88.44	0.015
	Consumption (BW^0.75^)										
CP	8.394	7.928	8.784	8.971	7.738 B	9.301 A	NS	*	NS	-	0.498
NDF	59.999	52.695	55.34	54.12	54.09 B	56.983 A	NS	*	NS	-	3.230
EE	2.425 d	3.382 c	4.756 b	5.81 a	3.862 B	4.322 A	*	*	NS	95.79	0.259
NFC	14.743 a	12.08 b	11.61 bc	9.89 c	10.62 B	13.545 A	*	*	*	88.92	0.690

* Significant effect (*p* < 0.05); NS—not significant (*p* > 0.05) Student’s *t*-test; Means in the same row followed by different uppercase and lowercase letters, respectively, differ from each other (*p* < 0.05) by Tukey’s test; LR—the less rainy season; WS—the wettest season; R^2^ = determination coefficient; SEM—standard error of the mean; CP = Crude protein; NDF = neutral detergent fiber; EE = ether extract-y = 0.32 + 0.43x, y = 0.05 + 0.07x, y = 2.60 + 3.41x; NFC = non fiber nitrogen-y = 1.71 – 0.50x, y = 0.283 – 0.091x, y = 14.02 – 4.43x.

**Table 5 animals-13-00934-t005:** Intake of non-fiber carbohydrates under the effect of interaction between treatment and period in buffaloes in the Amazon eastern.

Period	Inclusion PKC				*p* ^1^	R^2^	SEM ^2^
	0	0.25	0.5	1.0			
WS ^3^	16.0 aA	13.1 bA	13.1 bA	12.0 bA	*	73.6	0.506
LR ^4^	13.5 aB	11.1 bB	10.1 bB	7.8 cB	*	96.2	0.506
*p*	*	*	*	*			

^1^*p*-value; ^2^ Standard error of the mean; ^3^ Rainiest season; ^4^ Less rainy season; R^2^ = determination coefficient; * Significant effect (*p* < 0.05); Means in the same row followed by different uppercase and lowercase letters, respectively, are significant (*p* < 0.05) by Tukey’s test; Equations: WS-y = 15.07 − 3.47x; LR-y = 12.99 − 5.41x.

**Table 6 animals-13-00934-t006:** Equations for ruminal degradation, colonization time (CT) and effective degradability (ED) of dry matter (DM), fiber in neutral detergent (NDF) and crude protein (CP) of palm kernel cake by buffaloes in the eastern Amazon.

Items	Equations ^1^		R^2^ %	CT (h:min)	ED (2%/h) %	ED (5%/h) %	ED (8%/h) %
Dry Matter							
TP0	*p* = 26.28 + 52.11 (1 − exp^−0.0709t^)	A	97.70	3:54	50.30	39.06	35.96
TP0.2	*p* = 27.08 + 52.97 (1 − exp^−0.0596t^)	A	96.10	4:46	50.59	39.10	35.89
TP0.5	*p* = 26.32 + 60.42 (1 − exp^−0.0367t^)	A	97.30	6:39	50.85	39.29	35.14
TP1	*p* = 22.87 + 52.77 (1 − exp^−0.0316t^)	B	98.60	7:04	50.98	39.79	34.09
Fiber in Neutral Detergent							
TP0	*p* = 6.31 + 72.06 (1 − exp^−0.0227t^)	B	98.20	3:40	44.68	28.86	22.28
TP0.2	*p* = 8.66 + 84.38 (1 − exp^−0.0145t^)	A	92.00	6:43	44.12	27.63	21.60
TP0.5	*p* = 6.23 + 70.97 (1 − exp^−0.0236t^)	B	98.30	3:33	44.68	29.01	22.42
TP1	*p* = 5.61 + 75.08 (1 − exp^−0.0220t^)	B	97.80	3:16	44.97	28.57	21.82
Crude Protein							
TP0	*p* = 39.05 + 60.12 (1 − exp^−0.0193t^)	A	98.10	1:20	68.58	55.80	50.74
TP0.2	*p* = 36.82 + 64.30 (1 − exp^−0.0196t^)	A	98.40	0:30	68.72	54.99	49.52
TP0.5	*p* = 26.32 + 60.42 (1 − exp^−0.0367t^)	A	98.30	0:15	69.14	55.14	49.78
TP1	*p* = 22.87 + 52.77 (1 − exp^−0.0316t^)	A	98.40	0:03	69.69	56.03	50.92

^1^ Orskov and McDonald’s model (1979). *p* = Amount of substrate degraded in time (t); CT = Colonization time; R^2^ = determination coefficient; ED = Effective degradability; Equations accompanied by the same capital letters in the same column are identical by the parameter identity test at 5% probability of type I error [26]. The highest supplementation level (PKC1) differed from the others in the behavior and distance from the DM degradation curves of palm kernel cake (*p* < 0.05).

**Table 7 animals-13-00934-t007:** Equations for ruminal degradation, colonization time (CT), effective degradability (ED) of dry matter (DM), fiber in neutral detergent (NDF) and crude protein (CP) of Marandu grass as a function of increasing levels of palm kernel cake for buffaloes in the eastern Amazon.

Items	Equations ^1^		R^2^ %	CT (h:min)	ED (2%/h) %	ED (5%/h) %	ED (8%/h) %
Dry matter							
TP0	*p* = 16.64 + 73.96 (1 − exp^−0.0265t^)	A	99.40	02:30	50.83	35.93	29.31
TP0.2	*p* = 17.43 + 71.66 (1 − exp^−0.0273t^)	A	99.60	02:27	50.76	35.42	30.22
TP0.5	*p* = 18.60 + 82.04 (1 − exp^−0.0226t^)	A	98.90	02:07	50.44	35.20	29.83
TP1	*p* = 16.20 + 77.93 (1 − exp^−0.0254t^)	A	99.60	01:09	50.16	34.59	28.81
Fiber in neutral detergent							
TP0	*p* = 1.17 + 79.23 (1 − exp^−0.0230t^)	B	99.20	00:37	43.62	26.20	18.91
TP0.2	*p* = 1.64 + 78.76 (1 − exp^−0.0238t^)	AB	99.30	00:41	44.45	26.05	19.70
TP0.5	*p* = 1.48 + 81.01 (1 − exp^−0.0243t^)	AB	99.20	00:48	44.21	26.49	19.15
TP1.0	*p* = 1.71 + 78.35 (1 − exp^−0.0258t^)	A	99.50	00:51	45.15	27.75	20.30
Crude Protein							
TP0	*p* = 39.89 + 28.64 (1 − exp^−0.0518t^)	B	97.10	00:40	60.56	54.47	51.15
TP0.2	*p* = 37.04 + 30.73 (1 − exp^−0.0570t^)	A	97.30	02:08	59.85	53.49	49.90
TP0.5	*p* = 36.22 + 30.88 (1 − exp^−0.0575t^)	A	97.70	02:18	59.57	52.74	48.91
TP1.0	*p* = 37.15 + 30.80 (1 − exp^−0.0605t^)	A	95.90	02:27	60.31	54.03	50.42

^1^ Orskov and McDonald’s Model (1979). *p* = Amount of substrate degraded in time (t); R^2^ = determination coefficient; CT = Colonization time; ED = Effective Degradability. The groups TP0.00; TP0.25; TP0.50 and TP1.00 correspond to treatments of 0.0%, 0.25%, 0.50% and 1.0% of palm kernel cake in relation to the BW of the buffaloes; Equations accompanied by equal capital letters in the same column are identical by the parameter identity test at 5% probability of type I error [26].

**Table 8 animals-13-00934-t008:** Equations of ruminal degradation, colonization time (CT), effective degradability (ED) of dry matter (DM), neutral detergent fiber (NDF) and crude protein (CP) of wheat bran as a function of increasing levels of palm kernel cake for buffaloes in the eastern Amazon.

Items	Equations ^1^		R^2^ %	CT (h:min)	ED (2%/h) %	ED (5%/h) %	ED (8%/h) %
Dry Matter							
TP0	*p* = 48.77 + 30.31 (1 − exp^−0.0886t^)	C	97.60	0:28	73.50	68.15	64.70
TP0.2	*p* = 57.19 + 23.84 (1 − exp^−0.0368t^)	B	97.40	2:00	72.64	67.30	64.70
TP0.5	*p* = 59.62 + 24.85 (1 − exp^−0.0358t^)	A	82.10	2:09	75.57	70.00	67.31
TP1	*p* = 59.90 + 23.18 (1 − exp^−0.0380t^)	A	92.90	2:09	75.10	69.92	67.37
Neutral detergent fiber							
TP0	*p* = 1.32 + 89.99 (1 − exp^−0.0492t^)	A	99.00	0:17	65.33	45.99	35.62
TP0.2	*p* = 1.85 + 89.98 (1 − exp^−0.0452t^)	A	98.70	0:27	64.24	44.58	34.35
TP0.5	*p* = 1.12 + 89.99 (1 − exp^−0.0487t^)	A	99.30	0:15	64.92	45.53	35.18
TP1	*p* = 1.96 + 89.99 (1 − exp^−0.0484t^)	A	99.30	0:26	65.65	46.24	35.90
Crude Protein							
TP0	*p* = 37.54 + 55.44 (1 − exp^−0.1488t^)	A	98.30	0:14	86.41	79.04	73.60
TP0.2	*p* = 27.19 + 64.46 (1 − exp^−0.1991t^)	B	98.40	1:03	85.78	78.72	73.19
TP0.5	*p* = 17.18 + 74.82 (1 − exp^−0.1898t^)	C	98.50	1:52	84.91	76.44	69.85
TP1	*p* = 21.95 + 89.99 (1 − exp^−0.2088t^)	D	98.40	2:34	84.09	74.57	67.03

^1^ Orskov and McDonald’s Model (1979). *p* = Amount of substrate degraded in time (t); R^2^ = determination coefficient; CT = Colonization time; ED = Effective Degradability. The groups TP0.00; TP0.25; TP0.50 and TP1.00 correspond to the treatments 0.0%, 0.25%, 0.50% and 1.0% of palm kernel cake in relation to the BW of the buffaloes; Equations accompanied by equal capital letters in the same column are identical by the parameter identity test at 5% probability of type I error [26].

**Table 9 animals-13-00934-t009:** Performance of female buffaloes under intermittent grazing of Marandu grass supplemented with different levels of palm kernel cake.

Variables	Treatments				R^2^	SEM ^1^	CV ^2^	*p*
	TP0	TP0.25	TP0.5	TP1				
LR ^7^								
IW ^3^ (Kg)	508.0	494.0	502.0	508.0	-	57.27	0.11	0.94
FW ^4^ (Kg)	606.0	580.0	595.0	596.0	-	48.66	0.08	0.89
TAG ^5^ (Kg)	98.0	86.0	93.0	88.0	-	19.40	0.21	0.57
ADG ^6^ (Kg)	0.544	0.478	0.517	0.489	-	0.11	0.21	0.57
WS ^8^								
IW (Kg)	612.0	604.0	603.0	601.0	-	57.01	0.09	0.76
FW (Kg)	673.0	656.0	629.0	639.0	-	56.72	0.09	0.29
TAG (Kg)	61.00	52.0	26.0	38.0	-	27.19	0.61	0.12
ADG (Kg)	0.339	0.289	0.144	0.211	-	0.15	0.61	0.12

^1^ Standard error of the mean; ^2^ Coefficient of variation; ^3^ Initial weight; ^4^ Final weight; ^5^ Total average gain (180 days); ^6^ ADG = Average daily gain; ^7^ Less rainy season; ^8^ Wetter season; R^2^ = determination coefficient; *p* = *p*-value.

## Data Availability

The data presented in this study are available on request from the corresponding author.

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
