# Peer review of "Effect of Palm Kernel Cake Supplementation on Voluntary Feed Intake, In Situ Rumen Degradability and Performance in Buffaloes in the Eastern Amazon"

_animals, 2023, doi:10.3390/ani13050934_

Round 1
Reviewer 1 Report (Previous Reviewer 2)
No comments. The manuscript has been improved.
Author Response
Please see the attachment!
Requests were accepted. We appreciate the contributions. Here's the new version.

Reviewer 2 Report (Previous Reviewer 1)
General remarks
Dear authors,
I have evaluated the revised version of the manuscript identified as animals-2087078. Compared to the first version, the manuscript has been substantially improved, especially in the linguistic form. I positively evaluate the change of title compared to the previous version, as well as the changes made during the description of the results. However, in my opinion, the manuscript needs further modifications to be eligible for publication. In addition, some perplexities come to my mind. I have listed my concerns and suggestions below. I wish you good work.
Specific comments
Keywords: I have nothing to complain about regarding the choice of keywords. However, I believe the authors can modify the sequence, placing the by-product under study in the foreground rather than the target species. Thanks.
References: as per journal standards, references should be placed in the text in Vancouver style, i.e., ordered numerically according to their first appearance in the text. The same goes for the reference list, where cited manuscripts should be listed according to the order of their appearance in the text. In the current form of the manuscript, different styles are mixed, while the list of references is out of date (missing some citations) and does not follow the correct order. Authors are requested to conform the style to the journal's standard, correctly updating the list of references. Thanks.
L 124 (and through the text): in my opinion, the authors use the term "mineral matter" as a synonym for ashes. This is probably the case in their native language, but not otherwise. I invite the authors to clarify this aspect and modifying accordingly. Thanks.
L 127-129: the authors estimated, by difference, the content of non-fibrous carbohydrates using the equation proposed by Hall (which has been omitted from the list of references!!!). However, according to the aforementioned equation, ashes are used as a known term rather than mineral matter (see the previous comment) but, above all, urea derived from CP and urea are requested. Therefore, for the authors to be able to use this equation, I expect that the urea (in both forms described by the equation) has been dosed. But this, if true, has not been indicated in the analytical methods, nor even more so in the results in the table (the tables should always be self-explanatory and allow the reader to reconstruct the estimated data). In fairness, however, I believe the authors calculated the non-structural carbohydrates (i.e., NSC; see the 2001 NRC guide for a comprehensive elucidation of the differences between NSCs and NFC) according to the following equation:
NSC= 100 - (CP% + EE% + NDF%+ ashes) (NRC, 2001).
Clearly, by making the calculation backward based on the values ​​in the table with the NRC equation, we obtain exactly the value reported by the authors for NSC (not for NFC). Ultimately, the authors are requested to make the necessary modifications and corrections if they have carried out the evaluation of the NFC as indicated by Hall, or vice versa (as I suppose) indicating that the NSC has been estimated and, correctly, modify the text. Thank you for your patience.
L 167-207: an accurate bibliographic reference is required for these methods. Thanks.
L 277-335: the authors declare the significance of the differences found regarding several rumen degradability parameters. However, this is not evident from the values shown in the tables. Accordingly, I ask the authors if they could update the tables. Thanks
Author Response
Please see the attachment!
Requests were accepted. We appreciate the contributions. Here's the new version.

Round 2
Reviewer 2 Report (Previous Reviewer 1)
Dear authors,
I revised the latest version of the manuscript identified as animals-2087078. On the basis of the performed changes, I have no doubts about the scientific robustness of the manuscript. Therefore, I will suggest the paper's publication.
Congratulations
This manuscript is a resubmission of an earlier submission. The following is a list of the peer review reports and author responses from that submission.
Round 1
Reviewer 1 Report
General remark
Dear authors,
I have reviewed your manuscript identified as animals-2007825 (Intake, ruminal degradability and performance in buffaloes supplemented with palm kernel pie in the Eastern Amazon). I find the manuscript interesting, especially from the point of view of recycling agro-industrial by-products. This need, among other things, is particularly felt in the Amazon area, where deforestation to produce feed, among others, is a contingent problem. Therefore, the manuscript deserves to be published, in my opinion. However, I have noted some inaccuracies that deserve to be addressed. My doubts are detailed below. I hope my suggestions will improve the manuscript. Good luck.
Specific comment
L 42: Please, use “different” instead “distant”. Thanks.
L 47: in my opinion, the term "in situ" is not a sufficiently explanatory keyword. At the same time, is unclear to me what the authors mean by "offering". Please improve this part so that the manuscript can be properly identified through the keywords. Thanks.
L 56-58: in my opinion, the statement needs to be referenced. In this regard, I suggest doi.org/10.3390/ani9110918. Thanks.
L 59-62: I believe the sentence can be improved in terms of clarity. I think it should be specified that the palm kernel cake has a low cost, making it interesting. In addition, the statement "the studies show contents" can be omitted in favor of more concise language. Thanks.
L 64-67: I invite the authors to better structure the experimental hypothesis.
L 78: in my opinion, the Afi climate type can be explained. Thanks.
L 82: I think “feed intake” is more appropriate to “consumption”. Thanks.
L 82-90: the statements in this section are a bit confusing. First, it would be necessary to specify the crossbreed type, as well as the group's description, which could be performed more concisely, avoiding redundancies. Again, I do not agree that the repetition of the experiment in two different seasons can be considered a repeated measure since the same animals are not used (if I have not misunderstood).
L 91: check if “analyzes” is correct. Thanks.
L 91-114: the description of the proximate analysis methods (reported in Table 1 and elsewhere) has not been made in the text. The authors are requested to update it. Thanks.
L 93-94: unless strictly necessary, the authors could use a single taxonomic term to indicate the Marandu grass. Thanks.
L 100 (and throughout the text): what do the authors mean by MS? Maybe dry matter. In addition, NCF correspond to NFC (see table 2)? The authors are requested to standardize the acronyms in the text. Thanks.
L 117 (and throughout the text): please, use “feed” instead of “food” (it refers to human intention). Thanks.
L 121 (and throughout the text): subscripts should be used in writing chemical formulas. Thanks.
L 164: how the NSC was calculated should be indicated. Thanks.
L 169 (and throughout the text): the square should be put as a superscript. Thanks.
L 202: what is harvesting time? Please, explain. Thanks.
L 305: ADE is not a usual form. What do the authors mean by daily earnings?
L 321: please, replace “superiority in the” with “higher” and “in relation” with “compared”. Thanks.
L 336-340: what the authors correctly stated about the relationship between the energy density of the diet and the fiber degradability is also highlighted in a recent study (doi.org/10.3390/fermentation8080351) specifically focused on buffalo. Therefore, the authors could recall it, adding it to the references. Thanks.
L 367-369: in my opinion, the conclusions cannot be reduced to a mere summary of the results. I invite the authors to reformulate the conclusions by briefly recalling the experimental hypotheses and based on these, highlighting the results obtained and possible future research on the subject. Thanks.
Author Response
The suggestions were accepted.
The manuscript was proofread by a native.
We would like to resend the new version.

Reviewer 2 Report
The authors evaluated the supplementation of Palm Kernel Cake (PKC) on daily weight gains of buffaloes grazing Brachiaria brizantha. In general terms, the experiment is sound, however this manuscript needs substantial improvements before being ready for publication. Obviously this manuscript has not been revised by a native English-speaking person. I also recommend the authors a meticulous re-writing of the manuscript. There are too many editorial flaws that make this manuscript difficult to read and understand.
General comments
There is an extensive number of typos (L139), grammatical errors (L105; 150-153), and non-sense writing (L32; L85; L105; L139; L150).
There is no description of the statistical model utilized. With the information provided I cannot determine whether the statistical analysis was performed properly.
Please explain what is the difference between mixed-breed (L83) and crossbred (L89) buffaloes.
There are no descriptions of the mills used for processing samples. Given this is an in situ study with pore bags, this information is relevant.
There is no description of how Fraction B was determined. This is not acceptable.
The authors are mixing portuguese and English elsewhere (e.g., L370-377; suplemento in Table 2 and PV in Table 3). Obviously this manuscript has not been revised by a native English-speaking person.
There is no description of "P" or "G" in Table 3.
The formating of the paper is not ready/appropriate for publication (see L220-228).
I noticed the use of "," for decimals. I'm not sure what is the policy in Animals for that. In the US "," is used for thousands. Maybe the editor can clarify this.
Most tables are extremely, and unnecesarily, bussy. Please take out equations and R2.
Specific comments:
L32: Eastern Amazon?
L54: alternative to what?
L59: please, add the scientific name of palm kernel.
L86: months of age.
L121: kg DM feces/day
L121: describe the bolus characteristics.
L137: "Fecal excretion (FE)"
L138: What is "pre-dried" to me they were dried or not, and "pre" is redundant. Please delete.
L139-140: I interpret you determined undigested ADF. Delete "insoluble"
L150: used for what purpose?
L150-153: There is no verb in this sentence.
L180: The model is not written properly.
L198: Statistical analysis is poorly described (e.g., there is no model).
Table 2: supplement and not suplemento.
Table 3. the title of this table must be re-written.
Table 4. Multiple things must be addressed in this table (i.e., no units, inconsistent format relative to Table 3, etc).
Table 6. It is hard to understand this table. There are no P values and there are 4 variables to be tested. Same for Tables 7 and 8. Also, the citation has a different format.
Author Response

(The authors gave the same response as above.)
